# The relationship between weight history and psychological health—Differences related to gender and weight loss patterns

**Franziska U. C. E. Jung**[1]\*, **Steffi G. Riedel-Heller**[1‡], **Claudia Luck-Sikorski**[2‡]

**1** Institute of Social Medicine, Occupational Health and Public Health, Leipzig University, Leipzig, Germany,
**2** SRH University of Applied Health Sciences, Gera, Germany

‡ These authors share last authorship on this work.
\* franziska.jung@medizin.uni-leipzig.de

## Abstract

### Background

The prevalence and burden of obesity continues to grow worldwide. Psychological comorbidities may not only influence quality of life, but may also hinder successful weight loss. The causality between excess weight and mental health issues is still not fully understood. The aim of the study was to investigate whetherweight history parameters, (ie.age of onset) are related to psychological comorbidities.

### Method

The data were derived from a representative telephone survey in Germany, collecting information on weight loss patterns and mental health outcomes among individuals with BMI>30kg/m$^2$. Overall, 787 participants were examined in terms of depressive symptoms (Patient Health Questionnaire, PHQ-9) and anxiety (Generalized Anxiety Disorder Questionnaire, GAD7). In addition, participants were asked about different aspects of their weight history (ie. weight loss patterns and trajectories) over the lifespan. The relationship between weight history and mental health was analyzed using multivariate statistics.

### Results

According to regression analyses, having had more weight loss attempts, a greater weight loss being desired and being a "weight maintainer" was associated with more symptoms of depression (p < 0.001), whereas a greater desired weight loss and being categorized as a "weight maintainer" was associated with more anxiety (p < 0.001). Moroever, the prevalence of depressive symptoms was significantly higher in male individuals who desire to lose more weight or had more weight loss attempts in the past.

### Conclusion

Gender-specific differences were observed in terms of weight history parameters, as well as mental health outcomes. Especially for men, weight loss patterns seem to be related to

**Data Availability Statement:** The data that support the findings are available on figshare: DOI 10.6084/m9.figshare.21581754.

**Funding:** This work was supported by the Federal Ministry of Education and Research (BMBF), Germany, FKZ: 01EO1501 and supported by Open Access Publishing Fund of Leipzig University. The funders had no role in study design, data collection and analysis, decision to publish, or preparation of the manuscript.

**Competing interests:** The authors have declared that no competing interests exist.

depressive symptoms. Concerning the overall results, it becomes clear that screening for weight history at the beginning of a multidisciplinary weight loss program in the context of gender-specific psychological comorbidities is important. The question remains why some aspects of weight history seem to be more important than others.

## Introduction

Overweight as well as obesity continue to show rising prevalence worldwide, especially in children and adolescents [1,2]. Currently, the coronavirus disease pandemic (COVID-19) has been associated with both poor mental health as well as unfavorable weight-related health behaviors especially in people with obesity [3]. Apart from a rising prevalence, research on obesity has also focused on different facets of weight history and their impact on comorbidities and overall health, especially in terms of widespreading consequences over the lifespan. According to the life course approach, numerous biological, psychosocial, and cognitive factors can have an independent, cumulative, and interacting impact on the likelihood of developing health problems as people age [4,5]. Especially due to the complexity of obesity, patient-centered care has shown to be important to treat obesity and its comorbidities [6]. So far, a clear definition of weight history does not exist, however, it has been described as an evaluation of "historical information on a patient's weight gain (or loss) pattern and trajectory" and may be useful to identify treatment options and health risks, for instance with regard to psychological health [7].

Epidemiological studies conclude that up to 23.2% of all women with obesity and 11.7% of all men with obesity are affected by depressive symptoms [8]. Obesity and mental disorders such as depression do not only share a high co-morbidity, but also a functional association in terms of biological pathways [8–10]. Previous studies suggest a bi-directional relationship [11]. In other words, depression may lead to obesity later on in life. On the other hand, obesity may also be a risk factor for developing symptoms of depression. In addition, it has been shown that weight management efforts are associated with long-term psychological improvements [12,13]. However, even if weight loss may have many favorable effects on health and overall well-being, weight loss patterns and trajecories over time may also be associated with negative consequences.

In general, duration of being overweight as well as time of onset has been shown to predict risks of mortality and morbidity, possibly due to long-term exposure [14]. Even if an early onset is not a preliminary factor for being obese as an adult, studies suggest that becoming obese during childhood is a risk factor for staying obese until adolescence or even adulthood [15]. So far, mental health issues have been identified as a potential consequence of childhood overweight and obesity, especially if individuals continue to have excess weight through adulthood [16]. Studies also suggest that early onset, and therefore extended duration of obesity and overweight determines the risk for other comorbidities, which may additionally increase the risk for depressive symptoms [17,18]. However, previous studies only investigated onsets in childhood or adolescence but do not consider further stages throughout adulthood (such as early or late adulthood) as a starting point for obesity. Apart from obesity onset, other aspects of weight history seem to be important as well.

The frequency of weight loss attempts may also influence psychological health, especially if constant attempts to lose weight do not lead to improvements of overall health and well-being, but result in feelings of guilt, poor self-esteem and depression. In addition to that, research has shown that weight loss cannot per se be associated with favorable psychological effects and

may increase depressive symptoms [19,20]. In addition, unrealistic weight loss goals or expectations can enhance these negative effects [21]. In the past, greater weight loss desire was significantly associated with higher frequency of mentally and physically unhealthy days and not meeting one's expectations with regard to weight loss may lead to psychological distress [22]. As a result, individuals's weight history may be characterized by certain patterns including weight fluctuations, which have also been related to negative psychological outcomes [23].

Only a few studies concentrated on possible gender differences. Research suggests a tendency, that psychological health is differently influenced by weight history parameters such as onset–depending on the gender of the participants [24,25]. A recent five-year follow-up study has shown that presence of mood disorder at baseline results in later weight gain in male but not in female participants, whereas BMI at baseline was associated with higher risk for mood disorder in female, but not in male individuals [26]. However, most studies that investigated the association between weight history and mental health focused on female participants only. In addition, studies that conducted research on weight history trajectories are limited to restricted age ranges in childhood or adolescence.

To the best of our knowledge, this is the first study that focuses on male and female participants, investigating possible gender differences with regard to weight history and psychological health. In this context, the study aims to investigate the link between weight history parameters (obesity onset, frequency of weight loss attempts, weight loss patterns, greatest weight loss) and mental health outcomes (depression and anxiety) in a broad sample of individuals with obesity.

## Method

### Sampling procedure

Telephone interviews were sampled by a large market research institute, specialized on health care research (Forsa Institute of Social Research and Statistical Analysis, Berlin, Germany). Overall, 2,191 target households from all states in Germany were randomly contacted (sampling period: January-February 2015). Participants were selected using random digital dialing and Kish selection grid. Overall, 60.0% of these households could not be included into analysis due to refusal, drop-out during the interview or other reasons. Respondents were then interviewed by trained research interviewers using computer-assisted telephone interviews (CATI). In order to ensure representativeness, demographic weighing was applied with regard to age, gender and education. Overall, a target sample size of n = 1,000 individuals participated in the assessment. The inclusion criteria included being at least 18 years old and having a BMI greater than 30 kg/m$^2$. Of the n = 1,000 interviews, 125 (12.5%) were excluded from further analyses due to missing data on a majority of variables that were focus of this study. Furthermore, 88 participants were excluded because they stated having had metabolic surgery, which may have biased results on anxiety and depression. The conduct of this study was given approval by the Ethics Committee of the University of Leipzig (Approval No 208-14-14042014). The institute Forsa was responsible for obtaining verbal informed consent of each participant during the interview.

### Instruments

**Covariates.** Sociodemographic and other measures such as gender, age and body mass index (BMI) were included as confounders in all analyses. Information on self-reported weight and height was used to calculate the participants' BMI (in kg/m$^2$) with respect to WHO standards [27]. If respondents refused to state their current weight for personal reasons, they were asked whether their weight lies within a certain range in order to categorize them to the correct

BMI-category [28]. Representability of the German general public was esured by demographic weighing (including age, gender, and education).

**Characteristics of weight history.** Participants were asked about different aspects of their weight history as done in previous studies [29]. This included a question about the onset of obesity ("How old were you when you developed obesity?"). Based on these answers, the onset was categorized into four different groups: 1. Onset during childhood (before the age of 12), onset during adolescence (age 13–19), onset during early adulthood (age 20–40) and onset during later adulthood (age 41 or older). In addition, participants were asked about the frequency of weight loss attempts during their entire life ("How many times have you tried to lose weight?", open question). The answers were categorized as follows due to the distribution of the data: category 1 (1–2 times), category 2 (3–5 times), category 3 (6–10 times) and category 4 (more than 10 times).

In addition to onset, weight history was analyzed with regard to the number of times participants aimed to reduce their weight and the maximum weight loss they have achieved so far. Participants were asked about their greatest weight loss and their weight before they lost it. If participants' current weight exceeds the weight after their weight loss, they were categorized as "non-maintainers" and if their current weight was equal or lower, they were categorized as "maintainers". Participants were also asked for their weight loss goal, defined as "desired weight loss". Based on this information and in order to compare this in relation to their current weight, the difference between current and desired weight was calculated in percentage. A cut-off of ten percent was used as a key point, since guidelines suggest a realistic and healthy weight loss of ≤10% [6,30].

**Psychological health outcomes.** Depressive symptoms were assessed using the German version of the Patient Health Questionnaire (PHQ-9), containing 9 items, that have to be rated on a four-point- response scale (0 = not at all; 3 = nearly every day) [31,32]. A cut-off score of 10 or greater can be interpreted as a depressive disorder. In our sample, the internal consistency was $\alpha = 0.754$.

Anxiety was assessed using the Generalized Anxiety Disorder questionnaire (GAD-7). It contains 7 items and answers are given using a four-point response scale (0 = not at all; 3 = nearly every day) [33,34]. According to the authors, a cut-off score of 10 or greater can be interpreted as an anxiety disorder. The internal consistency in this sample was $\alpha = 0.822$.

### Data analysis

All data were analyzed using STATA/SE 16.0 [35]. Categorical variables were analyzed using $Chi^2$-test, continuous data were analyzed using one-way ANOVA. In addition, p-values and effect sizes (Cohen's d or Cramer's V were applicable) are reported. ANOVAS were also used investigating the effect of weight history variables and gender on psychological health (post-hoc: Tukey). In addition, partial correlations were part of the statistical analysis in order to investigate whether mental health outcomes and weight history parameters are related. Multivariate analysis (regression models), further investigating this relationship, include two models: Model 1 (dependent variable: symptoms of depression) and Model 2 (dependent variable: symptoms of anxiety), controlling for age, gender and current BMI. For partial correlations, the duration of being obese was included as a control variable.

### Results

The overall sample included 787 participants and the majority of this sample was male (53%). Details on sociodemographic information can be found in Table 1. In this sample, 11.7% fulfilled the cut-off for clinical relevant symptoms of anxiety (GAD-7: ≥10). Similar results could

**Table 1. Sociodemographic information for the overall sample and separated by gender.**

| | Overall (787) | Male (n = 417) | Female (n = 370) | p-value & effect sizes |
|---|---|---|---|---|
| **Age** | 55.8 (14.7) | 54.0 (14.4) | 57.9 (14.9) | $p < 0.001$, d = -0.264 |
| **Depression, PHQ-9 (0–24)** | | | | |
| M(SD) | 5.5 (4.2) | 4.9 (4.0) | 6.1 (4.4) | $p < 0.001$, d = -0.277 |
| Depressive Disorder($\geq$ 10) | 121 (15.3%) | 41 (11.5%) | 73 (19.7%) | $p = 0.002$, V = 0.131 |
| **Anxiety, GAD-7 (0–21)** | | | | |
| M(SD) | 4.4 (4.0) | 4.0 (3.8) | 4.9 (4.3) | n.s. |
| Anxiety Disorder($\geq$ 10) | 92 (11.7%) | 39 (9.3%) | 53 (14.3%) | $p = 0.031$, V = 0.131 |
| **BMI (30–56), M(SD)** | 34.4 (4.0) | 34.3 (3.9) | 34.5 (4.2) | n.s. |
| **Obesity Class I** | 521 (66.2%) | 279 (66.9%) | 242 (65.4%) | |
| **Obesity Class II** | 199 (25.3%) | 107 (25.7%) | 92 (24.9%) | |
| **Obesity Class III** | 67 (8.5%) | 31 (7.4%) | 36 (9.7%) | |
| **Onset of Obesity (yrs)** | | | | n.s. |
| M(SD) | 28.6 (16.0) | 28.2 (16.0) | 28.9 (15.9) | |
| **Frequency of weight loss attempts** (1–100) | | | | |
| M(SD) | 11.5 (18.6) | 10.2 (19.1) | 13.0 (18.0) | $p = 0.037$, d = -0.480 |
| **Desired weight loss (in %), n = 787** | | | | |
| **< 10%** | 150 (19.1%) | 104 (25.1%) | 46 (12.5%) | $p < 0.001$, V = 0.160 |
| **$\geq$ 10%** | 634 (80.9%) | 311 (74.9%) | 323 (87.5%) | |
| **Greatest weight loss (in kg)** | | | | |
| M(SD) | 16.2 (11.1) | 16.0 (11.4) | 16.4 (10.8) | n.s. |
| **Weight maintenance** | | | | |
| Maintainer | 193 (24.5%) | 115 (27.6%) | 78 (21.1%) | $p = 0.035$, V = 0.075 |
| Non-maintainer | 594 (75.5%) | 302 (72.4%) | 292 (78.9%) | |

Note: Categorical variables were analyzed using $\text{Chi}^2$-test; continuous variables were analyzed using one-way ANOVA; PHQ-9 = Patient Health Questionnaire; GAD-7 = Generalized Anxiety Disorder; BMI = Body mass Index, n.s. = not significant, M = mean, SD = standard deviation, p = statistical significance, d = Cohen's d, V = Cramer's V.

be obtained for depression, here, 15.3% of participants met the PHQ-9cut-off criterion of 10 and above for depression. With regard to psychological health, associated variables and gender, female participants showed significantly more symptoms of depression ($p < 0.001$). Female participants in this sample reported significantly more attempts of losing weight ($p = 0.042$), are less likely to maintain a previous weight loss ($p = 0.041$) and are more likely to wish for weight loss goals greater than 10% ($p < 0.001$).

## Weight history and psychological health

The severity of psychological factors differed between weight history categories as well as female and male participants (Figs 1 and 2).

Fig 1 demonstrates the mean scores of either depressive symptoms and symptoms of anxiety in terms of onset and frequency of weight loss attempts. With regard to onset, no significant differences could be found for female and male participants and between scores of depression or anxiety. Only in terms of number of weight loss attempts, male participants in this sample show significant differences in terms of depressive symptoms ($F(3,413) = 5.33$, $p = 0.001$), especially when comparing male participants who reported more than 10 weight loss attempts versus 1–2 attempts ($p = 0.001$) and versus 3–5 attempts ($p = 0.019$).

Similarly, Fig 2 describes psychological symptoms with regard to weight maintenance and weight loss goals. For weight loss goals, again, significant differences could only be found in male participants and with regard to depressiveness ($F(1,413) = 4.02$, $p = 0.046$), even though

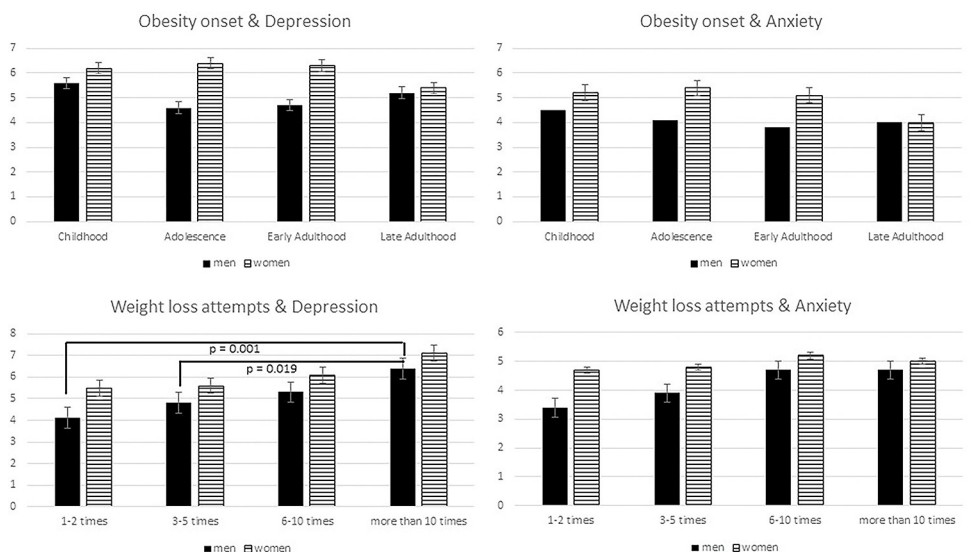

**Fig 1. Onset of obesity and weight loss attempts associated with psychological health for male and female participants (for depression: Mean PHQ-9 score, for anxiety mean GAD-7 score).**

on average, weight loss goals that exceed 10% were associated with more symptoms of depression in men (mean PHQ-9: 5.2 vs 4.3) and in women (mean PHQ-9: 6.2 vs. 5.2).

As a forth parameter, weight maintenance was analyzed. Interestingly, male participants characterized as "weight loss maintainer" show higher scores of symptoms of anxiety (mean GAD-7: 4.7 vs. 3.8) but not depressive symptoms (mean PHQ-9: 5.4 vs. 4.8) compared to "non-maintainers" ($F(1,415) = 5.48$, $p = 0.020$), as can be seen in Fig 2. For women, no significant differences could be obtained (mean PHQ-9: 6.2 vs. 6.1; mean GAD-7: 5.8 vs. 4.8).

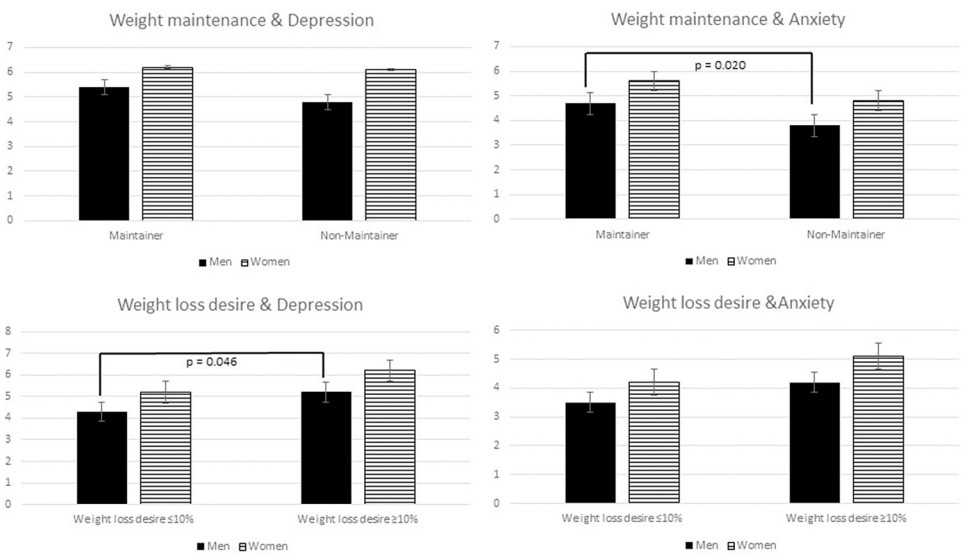

**Fig 2. Weight maintenance after greatest weight loss and desired weight loss associated with psychological health for male and female participants (for depression: Mean PHQ-9 score, for anxiety mean GAD-7 score).**

## Partial correlation (controlling for BMI, age and gender)

Results of partial correlations are summarized in Table 2. After controlling for age, BMI and gender, symptoms of depression were significantly correlated with anxiety, weight loss attempts and desired weight loss. In other words, the greater the number of weight loss attempts and the higher the desired weight loss, the more symptoms of depression can be found.

Anxiety was significantly correlated with desired weight loss and weight loss maintenance, meaning that greater symptoms of anxiety can be found in participants that have a higher desired weight loss goal and, interestingly, are categorized as weight maintainer.

## Regression analysis

In order to further investigate the influence of weight history factors on psychological health, multiple regression analysis was conducted (Table 3). Model 1 (dependent variable: depressive symptoms, PHQ-9) shows that having had more weight loss attempts, a greater weight loss being desired and being a "weight maintainer" can be associated with more symptoms of depression ($F(10,776) = 6.10$, $p < 0.001$, $R^2 = 0.071$, $Eta^2 = 0.074$). Model 2 (dependent variable: symptoms of anxiety, GAD-7) shows that a greater desired weight loss and being categorized as a "weight maintainer" can be linked to more anxiety ($F(10,776) = 4.82$, $p < 0.001$, $R^2 = 0.078$, $Eta^2 = 0.059$).

## Discussion

Early onset of obesity and overweight has been shown to act as predictors for morbidity, increasing the risk for comorbidities and premature mortality, even after controlling for body mass index during adulthood [36,37]. The aim of this study was to investigate whether certain characteristics of weight history, such as frequency of weight loss attempts, weight maintenance, desired weight loss and onset of obesity are related to differences in psychological health. Moreover, since most studies that cover this research area are based on female-only samples, another aim was to investigate whether gender-differences may influence this association.

**Table 2. Partial correlations for psychological and weight history variables (controlling for age, gender and BMI).**

|  | Depression (PHQ-9) | Anxiety (GAD-7) | Onset | Weight loss attempts | Desired weight loss | Greatest weight loss |
|---|---|---|---|---|---|---|
| **Depression (PHQ-9)** |  |  |  |  |  |  |
| **Anxiety (GAD-7)** | .769*** |  |  |  |  |  |
| **Onset**[1] | -.046 | -.035 |  |  |  |  |
| **Weight loss attempts**[1] | .123*** | .037 | -.095** |  |  |  |
| **Desired weight loss**[1] | .180*** | .151*** | .060 | .085* |  |  |
| **Greatest weight loss**[1] | .045 | -.006 | -.233*** | .091* | .066 |  |
| **Weight maintenance**[2] | -.053 | -.100** | .019 | .030 | .097** | .125*** |

Note

*** Significant at .001 alpha level

** Significant at .01 alpha level

* Significant at .05 alpha level

[1] = continuous

[2] = dichotomous (0 = maintainer, 1 = non-maintainer); PHQ-9 = Patient Health Questionnaire; GAD-7 = Generalized Anxiety Disorder.

**Table 3. Multiple regression analysis with depression (Model 1) and anxiety (Model 2) as dependent variables.**

|  | Model 1: Depression, PHQ-9 | Model 2: Anxiety, GAD-7 |
|---|---|---|
| Onset[1] (ref. late adulthood) |  |  |
| Childhood | -.589 | -.285 |
| Adolescence | -.643 | -.384 |
| Early adulthood | -.622 | -.382 |
| Frequency Weight Loss Attempts[2] | -.025** | .005 |
| Desired weight loss[2] | .094*** | .083*** |
| Greatest weight loss[2] | .003 | -.012 |
| Weight Maintenance (ref.: maintainer) | -.880* | -1.285** |
| Age | .003 | -.034* |
| Gender (ref.: male) | .892** | 1.031** |
| BMI | -.078 | -.117* |

Note

*** Significant at .001 alpha level

** Significant at .01 alpha level

* Significant at .05 alpha level; PHQ-9 = Patient Health Questionnaire; GAD-7 = Generalized Anxiety Disorder

[1] = categorical

[2] = continuous.

With regard to psychological health, the mean score of symptoms of anxiety in this sample is higher compared to the general population [38]. Similarly, depressive symptoms in this broad sample are greater compared to other studies including individuals with normal weight [39], but slightly lower compared to other samples [40,41]. In general, female individuals in our sample exhibited more symptoms of both depression and anxiety compared to their male counterparts. These findings are similar to previous research, indicating that gender may be an important mediator in the relationship between excess weight and psychological health [42]. With regard to weight history, female participants in this study report significantly more weight loss attempts, similar to what can be found in previous studies [43,44]. In addition, female participants are also more likely to have greater weight loss desires (i.e. want to reduce more weight relative to their current weight) and are less likely to maintain their weight loss compared to men. Again, these findings go align with the results of other studies that focus on gender differences in weight management [45–47].

The main focus of this study was to investigate whether weight history parameters are related to psychological health. Overall, regression analysis showed that the number of weight loss attempts, the desired weight loss as well as weight maintenance can be associated with symptoms of depression, whereas, desired weight loss and weight maintenance influence the severity of anxiety symptoms. In our study, significant effects can only be found for men. Even if it has been suggested that the burden of psychological co-morbidities is associated with the duration (and hence onset) of obesity and overweight [48], our results do not verify this assumption. In other words, the association between onset and psychological health (depression and anxiety) was not significant. For depressive symptoms, it has been shown that the earlier depression is diagnosed, the greater the BMI increase in later life [49]. Another study has found that the bi-directional association between obesity and depression was stronger for females in young adulthood than in late adolescence and that obese adolescents have a 40% greater risk of being depressed [50]. In contrast, studies investigating treatment-seeking samples suggest that age of onset of obesity may not be associated with increased risk of psychopathology or negatively affect psychological health in general [51,52], which may explain the results of the current study.

In our study, the prevalence of depressive symptoms was significantly higher in male individuals who desire to lose more weight compared to individuals that wish to achieve a weight loss up to 10% a recommended by medical guidelines. Having a weight loss goal that may be unrealistic or exceeds medical recommendations may increase disappointment, increase pressure to lose weight and therefore negatively influence psychological distress [22,53,54]. Especially the fact that only men were significantly affected is an interesting finding, as recent literature so far only suggests that higher weight loss goals may be negatively associated with mental health in women [22,55]. It underlines the importance of male participants with obesity as an understudied group. Additionally, this finding may also explain the link between number of weight loss attempts and depressive symptoms in men. In this context, higher frequency of weight loss attempts (i.e. more than 10 times) was significantly related to a higher score of depression in men, but not in women. Therefore, the reason for more frequent attempts may be explained by the greater weight loss desire.

In the current study, weight maintenance in men is also related to mental health, showing that symptoms of anxiety are more common among individuals that can be categorized as weight maintainers (i.e. being able to maintain or further reduce weight that has been lost in the past). This is similar to previous research showing that weight loss per se may not only be mentally satisfying, especially if the original weight loss goal was rather unrealistic to achieve as pointed out before. Achieved weight loss may–for example–increase the pressure to keep this weight loss and therefore negatively impact psychological health by increasing distress. Similarly, several studies indicate that weight loss or weight maintenance may be related to negatives consequences with regard to psychosocial and mental health issues [20,56,57].

Overall, the fact that male participants in this study were significantly affected by weight loss history parameters compared to female participants highlights the importance of including and separately analyzing results with particular attention to gender differences. Since the relationship between obesity and mental health in men has often been overlooked by researchers, eliminating additional variables and confounders such as somatic conditions, that may moderate this relationship in men and explain higher rates of mental health problems in some cases [58,59].

Data were based on self-reports, which may have biased the results for instance due to underreporting weight or incorrectly remembering the age of onset. Previous research demonstred that individuals tend to over- or underreport their own weight and height–depending on gender and age [60]. Another limitation is related to weight maintenance. From our data, we can only categorize individuals as maintainer or non-maintainer based on the information, that they have been able to maintain a weight loss that has been lost in the past. However, we are not able to deduce what happened between this weight loss and the time of study conductance. Individuals, categorized as weight maintainers, may still show (unfavorable) weight cycling, even if they managed to maintain or even reduce their weight.

## Conclusion

Psychological comorbidities and their role in determining well-being and weight status are of great relevance. Our studies reveals that depression and anxiety are related to weight maintenance, weight loss goals as well as attempts. Therefore, weight history characteristics should be included within clinical decision-making as part of multi-modal obesity management programs. The results of this study do not only underlie why these patient-specific parameters are essential, as the severity of mental health issues in individuals with obesity may depend on certain weight trajectories, there may also be differences for male and female patients which should not be left out of focus. Future research should also focus on male participants, in order to identify risk groups that may suffer from mental health problems related to weight history.

## Author Contributions

**Conceptualization:** Franziska U. C. E. Jung.

**Formal analysis:** Franziska U. C. E. Jung.

**Funding acquisition:** Claudia Luck-Sikorski.

**Investigation:** Franziska U. C. E. Jung.

**Methodology:** Franziska U. C. E. Jung, Claudia Luck-Sikorski.

**Project administration:** Claudia Luck-Sikorski.

**Supervision:** Steffi G. Riedel-Heller.

**Writing – original draft:** Franziska U. C. E. Jung.

**Writing – review & editing:** Steffi G. Riedel-Heller, Claudia Luck-Sikorski.

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
