## [Decision Letter · Decision Letter 0]

2 Nov 2022

PONE-D-22-11713The relationship between obesity, weight history and psychological health – are there differences related to gender?PLOS ONE

Dear Dr. Jung,

Thank you for submitting your manuscript to PLOS ONE. After careful consideration, we feel that it has merit but does not fully meet PLOS ONE’s publication criteria as it currently stands. Therefore, we invite you to submit a revised version of the manuscript that addresses the points raised during the review process.

 We think this is an important topic but the writing is not at all clear. Please follow the STROBE reporting guideline mentioned by the reviewer, and submit a checklist along with your resubmission. 

We look forward to receiving your revised manuscript.

Kind regards,

Meng Li

Academic Editor

PLOS ONE

“This work was supported by the Federal Ministry of Education and Research (BMBF), Germany, FKZ: 01EO1501 and supported by Open Access Publishing Fund of Leipzig University.”

“This study was supported by by the Federal Ministry of Education and Research (BMBF), Germany, FKZ: 01EO1501 as well as the Open Access Publishing Fund of Leipzig University.”

“This work was supported by the Federal Ministry of Education and Research (BMBF), Germany, FKZ: 01EO1501 and supported by Open Access Publishing Fund of Leipzig University.”

6. We note that you have indicated that data from this study are available upon request. PLOS only allows data to be available upon request if there are legal or ethical restrictions on sharing data publicly. For more information on unacceptable data access restrictions, please see http://journals.plos.org/plosone/s/data-availability#loc-unacceptable-data-access-restrictions.

7. Please note that in order to use the direct billing option the corresponding author must be affiliated with the chosen institute. Please either amend your manuscript to change the affiliation or corresponding author, or email us at plosone@plos.org with a request to remove this option.

Reviewers' comments:

Reviewer's Responses to Questions

**Comments to the Author**

1. Is the manuscript technically sound, and do the data support the conclusions?

Reviewer #1: Yes

2. Has the statistical analysis been performed appropriately and rigorously? 

Reviewer #1: Yes

3. Have the authors made all data underlying the findings in their manuscript fully available?

Reviewer #1: No

4. Is the manuscript presented in an intelligible fashion and written in standard English?

Reviewer #1: Yes

5. Review Comments to the Author

Reviewer #1: Reviewer comments PONE D 22 11713

This paper describes an observational cohort study of the associations between anxiety and depression scores and obesity onset, weight loss goals and weight loss attempts in adults with obesity. The study is potentially important and valuable and the authors have a great opportunity to make a major contribution to the filed with the large amount of data they have collected. However, I have a number of comments to make about the paper, offered in a constructive spirit.

From the outset, there is an overwhelming sense of vagueness to the paper that pervades every part of it. The title is vague and does not adequately convey what the hypothesis is. What is the study design? What are the specific outcomes? In the abstract, the methods aren’t at all clear, the aims aren’t consistent with the title and the results just don’t make sense. Why the F statistic in the abstract results? This is unconventional.

The authors need to revise STROBE (or similar) guidelines and adhere to these in revising their paper.

Also it is worth noting that the formatting is not of a sufficiently high standard – taking the abstract as but one example.

The introduction is not adequately focussed on the hypothesis in hand. Arguably, the first two paragraphs could be deleted conmpletely and the paper would look much more succinct and focussed. The consideration of childhood obesity “tracking” into adulthood is to me totally irrelevant to this paper and just distracts from a consideration of the frankly really interesting findings that people with obesity who have a desire to lose a greater amount of weight tend to have higher levels of anxiety and depression,

Stop referring to the cohort as “non-clinical” – it diminishes the credibility of the paper.

How was BMI measured by phone? Have the authors considered the challenges this poses in terms of the reliability of the data? Is tehre a precedent for large studies using self reported weight and height? If so, it would be good for the authors to demonstrate an awareness of this.

The term “weight history” is vague. So is psychological health. Be specific about what these are much earlier and throughout the paper.

The failure to include error bars in figures 1 and 2 is a major omission.

The paper needs to be much shorter and more focussed and specific, especially the discussion and the introduction. Then the authors will have an important contribution to the literature.

6. PLOS authors have the option to publish the peer review history of their article (what does this mean?). If published, this will include your full peer review and any attached files.

Reviewer #1: **Yes: **Francis M Finucane

---

## [Author Response · Author response to Decision Letter 0]

18 Nov 2022

Dear Prof. Li,

We would hereby like to re-submit our manuscript (PONE-D-22-11713).

First of all, we revised the manuscript according to the jounral requirements mentioned in your Decision letter:

1. Style requirements were adjusted

2. Details were added with regard to participant consent

3. With regard to the funding, the following applies: “The funders had no role in study design, data collection and analysis, decision to publish, or preparation of the manuscript.”

4. The funding-related text was removed from the manuscript. We apologize for this misunderstanding. In addition, due to the agreement between PLOS and my institution (Leipzig University), I had to change the funding information in the online system.

5. / 6. : the minimal anonymized data set will be available here: DOI 10.6084/m9.figshare.21581754

7. I am affiliated with Leipzig University. Please let me know if you need any proofs.

In addition, we thank the Reviewer for these helpful comments and revised the manuscript accordingly. Especially the main focus of the study was concretized by adding more information and defining the construct of “weight history” as this may have caused confusion. The manuscript was also revised with regard to STROBE guidelines and the figures and tables were adjusted. More information on this can be found in the file uploaded (Response to Reviewer).

The final version of this article was read and approved by all mentioned authors

We would be very grateful, if you could re-consider our manuscript for publication. If there are any further questions, please do not hesitate to contact us.

Yours sincerely,

Franziska Jung

---

## [Decision Letter · Decision Letter 1]

4 Jan 2023

PONE-D-22-11713R1The relationship between weight history and psychological health – Differences related to gender and weight loss patternsPLOS ONE

Dear Dr. Jung,

Thank you for submitting your manuscript to PLOS ONE. After careful consideration, we feel that it has merit but does not fully meet PLOS ONE’s publication criteria as it currently stands. Therefore, we invite you to submit a revised version of the manuscript that addresses the points raised during the review process.

One of the original reviewers recommended rejecting this paper and the other reviewer recommended major revision. Usually, I would reject a paper in this type of situation. However, I'd like to give this paper another chance. Please address ALL comments to the best you can. In your response to reviewers' comments letter, please quote the revised manuscript section instead of saying "revisions have been made".  This will make it easier for the reviewers as they do not have to read your response letter and the revised manuscript side by side. 

We look forward to receiving your revised manuscript.

Kind regards,

Meng Li

Academic Editor

PLOS ONE

Reviewers' comments:

Reviewer's Responses to Questions

**Comments to the Author**

1. If the authors have adequately addressed your comments raised in a previous round of review and you feel that this manuscript is now acceptable for publication, you may indicate that here to bypass the “Comments to the Author” section, enter your conflict of interest statement in the “Confidential to Editor” section, and submit your "Accept" recommendation.

Reviewer #1: (No Response)

Reviewer #2: (No Response)

2. Is the manuscript technically sound, and do the data support the conclusions?

Reviewer #1: Partly

Reviewer #2: Partly

3. Has the statistical analysis been performed appropriately and rigorously? 

Reviewer #1: Yes

Reviewer #2: Yes

4. Have the authors made all data underlying the findings in their manuscript fully available?

Reviewer #1: Yes

Reviewer #2: Yes

5. Is the manuscript presented in an intelligible fashion and written in standard English?

Reviewer #1: Yes

Reviewer #2: Yes

6. Review Comments to the Author

Reviewer #1: Having carefully revised the amended version of the manuscript and the responses to my previous comments, I think that the overall quality of the writing and the adequacy of the consideration of those comments fall below an acceptable level. The changes have been minimal. The consideration of, for example, the methodological limitations with self-reported weights and heights obtained by phone (comment/ response 5) has been inadequate. F statistics remain in the abstract.

Reviewer #2: The authors conducted a phone interview study collecting information on weight loss patterns and history and mental health outcomes among individuals with obesity. I have some comments and suggestions to enhance the clarity of the manuscript.

1. In the Abstract’s Methods section (line 22): Please specify the study design in the first sentence.

2. In the Abstract’s Methods section: Please include what analyses were done (the Results section mentions “regression analyses” for the first time. What regression analyses the authors used needs to be explained briefly in the Methods.

3. In the Abstract’s Conclusion section (line 35): The Results section in the abstract does not include any gender-specific findings. The conclusion seems to be irrelevant in the abstract. Please rewrite accordingly.

4. In the Introduction section (lines 98-99): Please define what “four stages of onset” and “variety of parameters” are upfront in the Introduction for a clearer understanding. The current objective can sound quite vague.

5. In the Method’s Sampling Procedure section: When was this data collection via phone interviews done and for how long (study date and period)?

6. In the Method’s Sampling Procedure section: Please indicate the age inclusion criteria of participants (e.g., Were they individuals of all ages? Or individuals of X years and older? Adults?)

7. In the Method’s Characteristics of Weight History section: Were the participants asked to count all the weight loss attempts “during their entire life (ever since the obesity onset)”? Please specify such information.

8. In the Method’s Data Analysis section: Please explain partial correlation and multi-variable regression models in more detail and their objectives (Currently, the authors only explain what was done for each analysis in the Results section. This information should be addressed upfront in the Methods).

9. In the Results (the use of “onset”): The authors mention that they calculated the duration by subtracting the age of onset from the age at the interview. However, it is not clear which value the authors used in the analyses (was the duration used in the analyses at all?).

a. Table 1: Does the variable “Onset of obesity” mean the age of onset of the duration of obesity?

b. Table 2: Similar here. What does the variable “Onset” mean?

c. In Regression Analysis section: It seems the models only include the age of onset (in 3 categories). Would it be meaningful to also include the duration of obesity in the models?

10. In the Discussion section (line 254): What does it mean by “clinical samples”? Those with obesity? Please specify.

11. In the Discussion section: Maybe also compare the current results with previous studies about the consistently higher prevalence of anxiety and depression in women regardless of their weight loss attempts, goals, and weight maintenance.

Thank you.

7. PLOS authors have the option to publish the peer review history of their article (what does this mean?). If published, this will include your full peer review and any attached files.

Reviewer #1: **Yes: **Francis Finucane

Reviewer #2: No

---

## [Author Response · Author response to Decision Letter 1]

26 Jan 2023

Reviewer Comment Reply

Reviewer 1 Having carefully revised the amended version of the manuscript and the responses to my previous comments, I think that the overall quality of the writing and the adequacy of the consideration of those comments fall below an acceptable level. The changes have been minimal. The consideration of, for example, the methodological limitations with self-reported weights and heights obtained by phone (comment/ response 5) has been inadequate. 

1. F statistics remain in the abstract.

2. From the outset, there is an overwhelming sense of vagueness to the paper that pervades every part of it. The title is vague and does not adequately convey what the hypothesis is. What is the study design? What are the specific outcomes? In the abstract, the methods aren’t at all clear, the aims aren’t consistent with the title and the results just don’t make sense. Why the F statistic in the abstract results? This is unconventional.

 We thank the reviewer for this comment and revised the abstract accordingly. 

Background

The prevalence and burden of obesity continues to grow worldwide. Psychological comorbidities may not only influence quality of life, but may also hinder successful weight loss. The causality between excess weight and mental health issues is still not fully understood. The aim of the study was to investigate whether weight history parameters, (ie.age of onset) are related to psychological comorbidities.

Method

The data were derived from a representative telephone survey in Germany, collecting information on weight loss patterns and mental health outcomes among individuals with BMI>30kg/m2. Overall, 787 participants were examined in terms of depressive symptoms (Patient Health Questionnaire, PHQ-9) and anxiety (Generalized Anxiety Disorder Questionnaire, GAD7). In addition, participants were asked about different aspects of their weight history (ie. weight loss patterns and trajectories) over the lifespan. The relationship between weight history and mental health was analyzed using multivariate statistics.

Results

According to regression analyses, having had more weight loss attempts, a greater weight loss being desired and being a “weight maintainer” was associated with more symptoms of depression (p < 0.001), whereas a greater desired weight loss and being categorized as a “weight maintainer” was associated with more anxiety (p < 0.001). Moroever, the prevalence of depressive symptoms was significantly higher in male individuals who desire to lose more weight or had more weight loss attempts in the past.

Conclusion

Gender-specific differences were observed in terms of weight history parameters, as well as mental health outcomes. Especially for men, weight loss patterns seem to be related to depressive symptoms. Concerning the overall results, it becomes clear that screening for weight history at the beginning of a multidisciplinary weight loss program in the context of gender-specific psychological comorbidities is important. The question remains why some aspects of weight history seem to be more important than others.

 3. The authors need to revise STROBE (or similar) guidelines and adhere to these in revising their paper. Also it is worth noting that the formatting is not of a sufficiently high standard – taking the abstract as but one example. We agree with the reviewer and revised the manuscript and abstract with regard to the format and the specifications given by reviewer 2.

 4. The introduction is not adequately focussed on the hypothesis in hand. Arguably, the first two paragraphs could be deleted conmpletely and the paper would look much more succinct and focussed. The consideration of childhood obesity “tracking” into adulthood is to me totally irrelevant to this paper and just distracts from a consideration of the frankly really interesting findings that people with obesity who have a desire to lose a greater amount of weight tend to have higher levels of anxiety and depression, We apologize for this misunderstanding. The consideration of childhood obesity tracking into adulthood was added as part of the introduction, because it reflects the main focus of this paper (“weight history”) and underlines the importance of tracking BMI and other weight-related parameters as part of successuful weight managament over the life course – as suggested by others (e.g. Berry et al., 2022; Kushner et al., 2020; Ekberg et al., 2012; Ryder et al., 2019). The finding that people with obesity who have a desire to lose a greater amount of weight tend to have higher levels of anxiety and depression was not the main focus of this study. However, we think that the structure of the introduction may have caused confusing, therefore we revised this section thoughtfully.

Please see changes throughout the manuscript, for example:

Page 3, line 70-77:

So far, mental health issues have been identified as a potential consequence of childhood overweight and obesity, especially if individuals continue to have excess weight through adulthood [18]. Studies also suggest that early onset, and therefore extended duration of obesity and overweight determines the risk for other comorbidities such as Type 2 diabetes and cardiovascular disease, which may additionally increase the risk for depressive symptoms [19,20]. However, previous studies did not clearly show how different timepoints of onset (ie. childhod vs. early adulthood) may lead to differences in severity of depression or anxiety across the lifespan. In addition, other aspects of weight history seem to be important as well.

Page 3, line 83-87:

In the past, greater weight loss desire was signficiantly associated with higher frequency of mentally and physically unhealthy days and not meeting one’s expectations with regard to weight loss may lead to psychological distress [24,25]. As a result, individuals’s weight history may be characterized by weight fluctuations, which have been associated with negative psychological outcomes, therefore, there may be differences between weight-maintainers and non-maintainers [26].

 5. Stop referring to the cohort as “non-clinical” – it diminishes the credibility of the paper.

 We apologize for this misunderstanding. The aim was to make clear that we did not recruit patients from a hospital or GP practice like other studies. In other words, participants in our study may or may not be under treatment at the time of the data collection. We now changed this according to the reviewers’ comment. For example on Page 4, line 100.

 6. How was BMI measured by phone? Have the authors considered the challenges this poses in terms of the reliability of the data? Is tehre a precedent for large studies using self reported weight and height? If so, it would be good for the authors to demonstrate an awareness of this.

 We apologize for this misunderstanding. BMI was not “measured” by phone. Participants were asked to state their weight and height and BMI was then calculated from their response according to the formula provided by the WHO. In order to avoid missing values, participants who refused to provide information on weight were given a specific weight range, being able to assign them to one of the BMI categories (see Benecke, 2003 for more information). We are aware of the issues associated with self-reported weight and height, therefore we added this to the limitation section.

Page 5, line 123-127:

Information on self-reported weight and height was used to calculate the participants' BMI (in kg/m2) with respect to WHO standards [30]. If respondents refused to state their current weight for personal reasons, they were asked whether their weight lies within a certain range in order to categorize them to the correct BMI-category [31] . Representability of the German general public was esured by demographic weighing (including age, gender, and education).

Page 13, line 310-312:

Data were based on self-reports, which may have biased the results for instance due to underreporting weight or incorrectly remembering the age of onset. Previous research demonstred that individuals tend to over- or underreport their own weight and height – depending on gender and age (60).

 7. The term “weight history” is vague. So is psychological health. Be specific about what these are much earlier and throughout the paper.

 We thank the reviewer for this comment. “Weight history” and “psychological health” have now been described in greater detail throughout the manuscript.

Page 2, line 50-57:

According to the life course approach, numerous biological, psychosocial, and cognitive factors can have an independent, cumulative, and interacting impact on the likelihood of developing health problems as people age [4,5]. Especially due to the complexity of obesity, patient-centered care has shown to be important to treat obesity and its comorbidities [6]. So far, a clear definition of weight history does not exist, however, it has been described as an evaluation of “historical information on a patient’s weight gain (or loss) pattern and trajectory” and may be useful to identify treatment options and health risks, for instance with regard to psychological health [7].

 8. The failure to include error bars in figures 1 and 2 is a major omission.

 We apologize for this mistake. Error bars have now been added to the figures.

 9. The paper needs to be much shorter and more focussed and specific, especially the discussion and the introduction. Then the authors will have an important contribution to the literature.

 We agree with the reviewer and revised the introduction and discussion section. 

Reviewer 2 1. In the Abstract’s Methods section (line 22): Please specify the study design in the first sentence.

2. In the Abstract’s Methods section: Please include what analyses were done (the Results section mentions “regression analyses” for the first time. What regression analyses the authors used needs to be explained briefly in the Methods. We thank the reviewer for this comment and revised this section:

The data were derived from a representative telephone survey in Germany, collecting information on weight loss patterns and mental health outcomes among individuals with BMI>30kg/m2. Overall, 787 participants were examined in terms of depressive symptoms (Patient Health Questionnaire, PHQ-9) and anxiety (Generalized Anxiety Disorder Questionnaire, GAD7). In addition, participants were asked about different aspects of their weight history (ie. weight loss patterns and trajectories) over the lifespan. The relationship between weight history and mental health was analyzed using multivariate statistics.

 3. In the Abstract’s Conclusion section (line 35): The Results section in the abstract does not include any gender-specific findings. The conclusion seems to be irrelevant in the abstract. Please rewrite accordingly. We agree with the reviewer and revised the conclusion, but also the results section

Results

According to regression analyses, having had more weight loss attempts, a greater weight loss being desired and being a “weight maintainer” was associated with more symptoms of depression (p < 0.001), whereas a greater desired weight loss and being categorized as a “weight maintainer” was associated with more anxiety (p < 0.001). Moroever, the prevalence of depressive symptoms was significantly higher in male individuals who desire to lose more weight or had more weight loss attempts in the past.

Conclusion

Gender-specific differences were observed in terms of weight history parameters, as well as mental health outcomes. Especially for men, weight loss patterns seem to be related to depressive symptoms. Concerning the overall results, it becomes clear that screening for weight history at the beginning of a multidisciplinary weight loss program in the context of gender-specific psychological comorbidities is important. The question remains why some aspects of weight history seem to be more important than others.

 4. In the Introduction section (lines 98-99): Please define what “four stages of onset” and “variety of parameters” are upfront in the Introduction for a clearer understanding. The current objective can sound quite vague. We agree with the reviewer that this may have caused confusion and clearly defined the objective of the study by re-phrasing this sentences.

Page 4, line 101-103

In this context, the study aims to investigate the link between weight history parameters (obesity onset, frequency of weight loss attempts, weight loss patterns, greatest weight loss) and mental health outcomes (depression and anxiety) in a broad sample of individuals with obesity.

The four stages of onset (childhood, adolescence, early and late adulthood) are further described in the method section (page 5).

Page 3, line 75-77

However, previous studies only investigated onsets in childhood or adolescence but do not consider further stages throughout adulthood (such as early or late adulthood) as a starting point for obesity.

 5. In the Method’s Sampling Procedure section: When was this data collection via phone interviews done and for how long (study date and period)? We thank the reviewer for this comment and added this information to the manuscript.

Page 4, line 107-108

Overall, 2,191 target households from all states in Germany were randomly contacted (sampling period: January-February 2015).

 6. In the Method’s Sampling Procedure section: Please indicate the age inclusion criteria of participants (e.g., Were they individuals of all ages? Or individuals of X years and older? Adults?) Again, we thank the reviewer for this comment and added this information to the method section.

Page 5, line 113-114

The inclusion criteria included being at least 18 years old and having a BMI greater than 30 kg/m2.

 7. In the Method’s Characteristics of Weight History section: Were the participants asked to count all the weight loss attempts “during their entire life (ever since the obesity onset)”? Please specify such information. In consultation with a native speaker, we re-phrased the translation of this item. The question related to the weight loss attempts during their entire life and participants were given an answering scheme including the categories (1-2x, 3-5x, 6-10x), therefore, this was not an open question. Please see changes on page 6 (line 137-140).

In addition, participants were asked about the frequency of weight loss attempts during their entire life (“How many times have you tried to lose weight?”, open question). The answers were categorized as follows due to the distribution of the data: category 1 (1-2 times), category 2 (3-5 times), category 3 (6-10 times) and category 4 (more than 10 times).

 8. In the Method’s Data Analysis section: Please explain partial correlation and multi-variable regression models in more detail and their objectives (Currently, the authors only explain what was done for each analysis in the Results section. This information should be addressed upfront in the Methods).

 We agree with the reviewer and added more information on the data analysis section

Page 7, line 166-170

In addition, partial correlations were part of the statistical analysis in order to investigate whether mental health outcomes and weight history parameters are related. Multivariate analysis (regression models), further investigating this relationship, include two models: Model 1 (dependent variable: symptoms of depression) and Model 2 (dependent variable: symptoms of anxiety), controlling for age, gender and current BMI. For partial correlations, the duration of being obese was included as a control variable.

 9. In the Results (the use of “onset”): The authors mention that they calculated the duration by subtracting the age of onset from the age at the interview. However, it is not clear which value the authors used in the analyses (was the duration used in the analyses at all?).

a. Table 1: Does the variable “Onset of obesity” mean the age of onset of the duration of obesity?

b. Table 2: Similar here. What does the variable “Onset” mean?

c. In Regression Analysis section: It seems the models only include the age of onset (in 3 categories). Would it be meaningful to also include the duration of obesity in the models?

 We apologize for this mistake, the duration was not part of the current analyses (it was skipped after previous revision, because one cannot know what happened in between, e.g. change from obesity to overweight or normalweight). Therefore this sentences on page 5 was deleted from the manuscript.

a. “Onest of obesity” in Tab. 1 means the age at which a BMI over 30 was first observed/reached. Please see details in the method section (Item obesity onset: How old were you when you developed obesity”?)

b. Same here (therefore we indicated it as a continuous variable in Tab. 2)

c. Tab. 3 includes three age categories, because the forth category (late adulthood) was used as the reference category in this regression analyses. We therefore added this information to Tab. 3. If the reviewer thinks that it would make our results more comprehensible, it would also be possible to add another table with onset as a continuous variable instead. However, we did so due to the distribution of the data and in order to make it comparable to other studies that differentiate between age categories rather than using onset as a continous variable.

 10. In the Discussion section (line 254): What does it mean by “clinical samples”? Those with obesity? Please specify. By “Clinical samples” we mean samples derived from intervention studies or data collected in clinical settings, such as hospitals. It was not the focus of our study to ask participants whether they were currently under treatment, therefore we wanted to explain why results with regard to depressive symptoms in our sample are different to other samples that included patients with obesity, recruited in health care settings. 

As this was also mentioned by Reviewer 1, be deleted this in order to avoid any misunderstandings (please see page 11, line 260-261).

 11. In the Discussion section: Maybe also compare the current results with previous studies about the consistently higher prevalence of anxiety and depression in women regardless of their weight loss attempts, goals, and weight maintenance. We agree with the reviewer and think that it is important to mention this within the discussion section eventhough the main focus was on the relationship between weight history and mental health.

The following has been added to the manuscript:

Page 11, line 260-264

In general, female individuals in our sample exhibited more symptoms of both depression and anxiety compared to their male counterparts. These findings are similar to previous research , indicating that gender may be an important mediator in the relationship between excess weight and psychological health [45].

---

## [Editor Report · Decision Letter 2]

1 Feb 2023

The relationship between weight history and psychological health – Differences related to gender and weight loss patterns

PONE-D-22-11713R2

Dear Dr. Jung,

We’re pleased to inform you that your manuscript has been judged scientifically suitable for publication and will be formally accepted for publication once it meets all outstanding technical requirements.

Kind regards,

Meng Li

Academic Editor

PLOS ONE
---

## [Editor Report · Acceptance letter]

3 Feb 2023

PONE-D-22-11713R2 

The relationship between weight history and psychological health – Differences related to gender and weight loss patterns 

Dear Dr. Leipzig:

I'm pleased to inform you that your manuscript has been deemed suitable for publication in PLOS ONE. Congratulations! Your manuscript is now with our production department. 

Kind regards, 

on behalf of

Dr. Meng Li 

Academic Editor

PLOS ONE